# Does Carrying a Rider Change Motor and Sensory Laterality in Horses?

**DOI:** 10.3390/ani12080992

**Published:** 2022-04-12

**Authors:** Sophie Schwarz, Isabell Marr, Kate Farmer, Katja Graf, Volker Stefanski, Konstanze Krueger

**Affiliations:** 1Behavioural Physiology of Farm Animals, University of Hohenheim, Garbenstr. 17, 70599 Hohenheim, Germany; sophie@schwarz-ffb.de (S.S.); isy-marr@web.de (I.M.); volker.stefanski@uni-hohenheim.de (V.S.); 2Department of Equine Economics, Faculty of Agriculture, Economics and Management, Nuertingen-Geislingen University, Neckarsteige 6-10, 72622 Nürtingen, Germany; katja.graf@yahoo.de; 3Centre for Social Learning & Cognitive Evolution, School of Psychology, University of St Andrews, St Andrews KY16 9JPh, UK; katefarmer74@gmail.com; 4Zoology/Evolutionary Biology, University of Regensburg, Universitätsstraße 31, 93053 Regensburg, Germany

**Keywords:** laterality, horse, rider, sensory laterality, motor laterality, novel object, side preference

## Abstract

**Simple Summary:**

Laterality, or one-sidedness, has been studied in many species, including horses, and has been linked to factors such as stress and emotionality. Today, although most horses are used for riding, the impact that carrying a rider has on their sensory (preferred side of sensory organs) and motor (preferred side of body usage) laterality has not been researched to date. In this study, 23 horses were tested to assess, firstly, motor laterality by observing which foreleg a horse would use to step over a pole and, secondly, sensory laterality by observing the preferred side of sensory organs when exposed to (a) an unknown person and (b) a novel object. All three experiments were conducted with and without a rider. The rider gave minimal aids and rode on a long rein to allow the horse free choice. The results of this preliminary study show that the strength of motor laterality (the number of times the preferred foreleg was used) increased when horses carried a rider but that sensory laterality did not change. This suggests that carrying a rider who is as passive as possible does not have an adverse effect on a horse’s stress levels and mental state.

**Abstract:**

Laterality in horses has been studied in recent decades. Although most horses are kept for riding purposes, there has been almost no research on how laterality may be affected by carrying a rider. In this study, 23 horses were tested for lateral preferences, both with and without a rider, in three different experiments. The rider gave minimal aids and rode on a long rein to allow the horse free choice. Firstly, motor laterality was assessed by observing forelimb preference when stepping over a pole. Secondly, sensory laterality was assessed by observing perceptual side preferences when the horse was confronted with (a) an unfamiliar person or (b) a novel object. After applying a generalised linear model, this preliminary study found that a rider increased the strength of motor laterality (*p* = 0.01) but did not affect sensory laterality (*p* = 0.8). This suggests that carrying a rider who is as passive as possible does not have an adverse effect on a horse’s stress levels and mental state.

## 1. Introduction

To date, the effect of a rider on a horse’s cerebral laterality has not been investigated, although the existence of cerebral laterality in lower vertebrates has been known for almost a century [1,2], and it is well established today that brain lateralisation is widespread amongst vertebrates. It has been suggested that laterality in vertebrates and lower vertebrates stems from a common ancestor [3]. Cerebral laterality comprises motor and sensory laterality and is the result of different specialisations in the brain hemispheres [4,5]. Depending on the type of information and the situation, processing is either predominantly in the left or the right hemisphere as summarised by [6], with information transferring from the sensory organs and limbs on one side of the body to the brain hemisphere on the opposite side [7]. One proposed distinction between the functions of the hemispheres in vertebrates is to describe the left hemisphere as instruction driven and the right hemisphere as stimulus driven [6].

Lateralisation is found in a large number of animals, and evidence strongly suggests that it has evolutionary advantages [8]. Laterality can vary in strength; that is, it can be expressed as anything from a weak tendency to an almost universal preference, and it has been suggested that a strongly lateralised brain has many advantages [9]. In chicks, strong brain asymmetry has been shown to increase the chances of survival, as the different specialisations of the hemispheres allow them to simultaneously search for food and look out for predators [10].

Another example of a lateralised prey species is the horse (*Equus caballus*), and, in general, horses are dependent on their ability to be almost permanently alert to possible threats so that they can react quickly and appropriately when they find themselves in danger [11]. To this end, horses have special sensory traits, such as laterally placed eyes, which are very sensitive to movement and allow an almost 360-degree visual field [12]. Laterality also plays a big part in their survival skills. The specialisation of the two hemispheres allows them to spend most of their day seeking and ingesting food while simultaneously moving with their groups and scanning their surroundings for predators [13].

Today, the majority of the worldwide horse population lives as domesticated animals in human care [14], and their brain asymmetry affects not only equestrian sport but also their behaviour with respect to handling and husbandry. Their laterality has become a focus of scientific research in recent decades, as it has been proposed that traditional left-sided handling might influence a horse’s lateralisation [15] and, therefore, its training [16]. Laterality has been connected with different factors. For example, motor bias may increase with age and training [17]; different breeds of horses may have different side preferences [18]; and laterality has been shown to be correlated with sex, with female horses showing right-sided motor laterality more frequently than males [15]. Furthermore, research has now found a connection between horses’ laterality and their welfare and stress levels, with motor and sensory laterality showing a significant left shift under stress, such as a change in housing and initial training [19]. This suggests that the right brain hemisphere dominates when horses are stressed or their welfare is compromised [19,20,21].

As the majority of horses are bred and used for riding purposes, whether or not a rider influences a horse’s laterality could have implications for training [22,23], as well as indicating when horses may be suffering from stress in training [19]. However, to date, there have been no studies comparing motor and sensory laterality with and without a rider. Therefore, this study aims to examine whether a rider sitting on a horse affects motor and sensory laterality by addressing the following three questions: (i) Does motor and/or sensory laterality change in strength? (i.e., the frequency with which a preferred limb or sensory organs are used); (ii) Does motor and/or sensory laterality change in direction? (i.e., does the preference change from right to left or vice versa); and (iii) Does the sex, age or breed of the horse influence the expression of motor and/or sensory laterality?

To investigate this, the motor and sensory laterality of 23 horses were tested in three different experiments, and each horse was tested an equal number of times, both with and without a known rider. We investigated motor laterality with a foreleg preference test and sensory laterality with a person test and a novel object test.

## 2. Materials and Methods

### 2.1. Locations and Circumstances

The experiments were conducted in May 2021 at ten different stables in the area of Tübingen in Germany. Testing at a number of different premises minimised factors specific to just one location and allowed the horses to be tested in their familiar surroundings, thus reducing stress [24]. The stables were of different sizes and management systems, but all the horses had daily turnout in social groups for at least two hours per day. At each stable, enclosed spaces, such as round pens and lunging circles with a sand surface of roughly 20 m diameter, were used as the test areas. It was ensured that there were no immediate distractions during the trials that might affect the horses’ motivation; for example, there were no other horses in the testing area, and there were no accessible feed sources, including grass.

### 2.2. Animals

A total of 23 horses were tested (Appendix A), and they were all in good physical condition and regularly checked by local vets. All horses were ridden and were trained from the ground between one and seven times a week, with a median of six days a week, for about one hour each day. There were 18 geldings and 5 mares, with a median age of 13, ranging from 4 to 29 years. They were of various breeds: 3 Thoroughbreds, 13 warmbloods, 5 ponies and 2 draught horses. Seventeen of the horses were predominantly handled from the left side, while the other six were handled equally from both sides. They were privately owned, and, apart from one horse showing stereotypies, none of them usually expressed atypical behaviour.

### 2.3. Experimenters and Their Tasks

All experimenters participated in the planning and conducting of the study. Before conducting the test, they were informed in detail about the theoretical background and the aims of the study, the experimental procedure and their tasks. Fifteen experimenters participated in the tests and took over the tasks of E1, E2 or E3 as appropriate, depending on the test location.

#### 2.3.1. Experimenter 1 (E1)

E1 (*n* = 11) was the person who usually rode the horse and was the “rider”. Each rider conducted the experiment with a median of two horses (Min = 1; Max = 4; Appendix A). They were all at least at an intermediate level of riding. Apart from one, all were right handed, so this was not considered a variable. All experiments were carried out an equal number of times both with and without a rider. Before each rider trial, E1 mounted, using a mounting block, from a randomly chosen side to make sure the side of mounting would not influence the results. E1 then rode the horse around the enclosed space for one round in each direction, with a random change of rein, finishing with a straight line, to neutralise any mounting effects. During the trials, E1 did not talk to the horse, pet the horse or intentionally influence the horse in any way. E1 sat up straight and did not use any weight aids that could affect the horse. To move off in walk, E1 was allowed to give leg aids for one second but remained passive thereafter (i.e., the reins had to be loose at equal lengths); otherwise, E1 sat passively and kept looking straight ahead in the direction that the horse was moving.

#### 2.3.2. Experimenter 2 (E2)

E2 (*n* = 9) led the horse from the ground and was unknown to the horse at the start of the experiments. Each Experimenter 2 conducted the experiment with a median of three horses (Min = 1; Max = 4). All E2s had experience in handling horses for a minimum of 5 years. The same E2 led the horse randomly from the left and the right, in both ridden and unridden conditions, to keep the results comparable. E2 used minimal force on the horse. To ask the horse to proceed in walk, they applied pressure on the lead rope for only a brief moment so as not to disrupt the horse’s balance. Otherwise, they were passive towards the horse.

#### 2.3.3. Experimenter 3 (E3)

Finally, E3 (*n* = 10) was the test person who was passed by the horse in the sensory laterality person test. E3 stood passively, upright, with legs together and hanging arms, between a feeding bucket and the horse. E3 looked straight ahead in the direction of the approaching horse, facing a predetermined point behind the starting point and ignoring the horse during its approach. The details of the experimental procedure are described below. E3 was also responsible for placing different objects in the test area for the sensory laterality object test and for documenting the test results.

### 2.4. Experiments and Procedure

#### 2.4.1. Experimental Procedure

Each day, trials for each of the three experiments under both ridden and unridden conditions were conducted, with a maximum total of 16 trials on any one day. The trials were spread over at least three days, and the order of the experiments was chosen randomly on each day to avoid serial effects (Figure 1). Furthermore, the order of the unridden and ridden trials was randomly picked each day to avoid serial effects and to randomise the effect of tacking up and untacking the horse. The horses wore their usual tack for the rider trials, with three wearing Western-style saddles and bridles and the rest English-style saddle and bridles. A halter was put on over the bridle so that E2 could lead the horse while E1 held the reins. For the unridden trials, E2 used only a halter and rope. For each trial in each experiment, it was documented whether the horse showed left or right motor or sensory laterality, depending on which forelimb or side of facial sensory organs that the horse preferred to use.

The data were primarily documented by a camera, placed at the experimental arena opposite to the starting point and in a straight line from the starting point to the place where either the pole had to be crossed, the persons had to be passed or the objects had to be approached. The individual data of the horses and data on the horses’ lateral responses were also recorded by E3 by hand on a pre-set experimental sheet for each horse and checked against the video recording. Each horse’s laterality was classified as left, right or not clearly lateral, and the side that the horse was led and whether or not the horse carried a rider was noted (Appendix A). In the foreleg preference test, motor laterality was determined by which foreleg the horse used to step over the pole. Sensory laterality was determined by the side of the sensory organs that the horse used to investigate the object (Appendix A) in the object test and the person (E3) in the person test.

#### 2.4.2. Motor Laterality Foreleg Preference Test

For this experiment, a single pole was placed on the ground in the middle of the testing area. E2 led the horse around the area in a random direction for each trial to minimise the effect of the direction and then led it up to the pole in a straight line (Figure 2). The horse was stopped right in front of the pole, and E2 ensured it was standing with its weight equally distributed over all four limbs so that there was no predisposition to use either foreleg first. Once the horse was balanced, E2 gave the signal to proceed in walk, with the first step being over the pole. In the rider trials, E1 also gave initiating leg aids for one second but remained passive thereafter. As the horse took its first step, the foreleg used to step over the pole was documented by E3 on a data sheet after each trial and checked by viewing the video recordings after the test (Appendix A). Then, the procedure was repeated. For each horse, 20 trials were conducted over three days, 10 with a rider and 10 without. Under each condition, E2 led five times from the left and five times from the right.

#### 2.4.3. Sensory Laterality Person Test

Sensory laterality was observed by noting the side that the horse chose to pass a person (E3) to reach a bucket with feed. E3 was unknown to the horses at the start of the experiment. Prior to the first trial of each experimental session, the horse was allowed to eat some pieces of carrots out of the bucket to create an association between the bucket and the reward and, thus, to increase its motivation. E3 then placed a bucket with feed eleven metres away from the horse and then took position one metre in front of the bucket (Figure 3). E2 then led the horse to the predetermined starting point so that the bucket, E3 and the horse were in a straight line and so that neither side would be shorter or easier.

E2 then let the horse off the rope and gave no further indications or instructions to the horse. In the ridden trials, E1 was allowed to give leg aids to proceed in walk for one second but sat passively thereafter and let the horse find its own way. E3 stood still, looking past the horse at a predetermined location at the opposite side of the arena and completely ignoring the horse, even when it approached them, so as not to influence the horse with subconscious signals. Once the horse made its way to the bucket, the side of the sensory organs closest to E3 as it went past was documented as the sensory laterality; i.e., when the horse passed E3 with their right eye, nostril and ear to E3, right-sided sensory laterality was noted, and when it passed E3 with the left sensory organs, left sensory laterality was noted. Again, E3 documented the side choices of the horse on a data sheet after each trial and checked the data with the video recording (Appendix A). As in the motor laterality test, 20 trials were conducted over at least three days, 10 with a rider and 10 without, and E2 randomly led from the left or right side under each condition.

#### 2.4.4. Sensory Laterality Object Test

Six novel objects of different colours, sizes and shapes were selected for each horse. A list of the objects is provided in the Appendix A. E3 was responsible for placing and replacing the novel objects in each trial while the horse was looking away. E2 positioned the horse at a predetermined starting point in the experimental area, fifteen metres away from the novel object, and then led the horse straight up to the object (Figure 4). E3 recorded which side of its face it used to investigate the object on a data sheet after each trial and checked the results by viewing the video recordings after the test. If the horse did not show any preferred side, the trial was recorded as unclear laterality. The procedure was repeated with E1 as a rider for three different objects and without E1 for the other three. E2 randomly alternated the side on which they led the horse under both conditions. E1 was allowed to give leg aids for one second to initiate walk but then remained passive, facing the object, throughout the trial.

### 2.5. Ethics Statement

The local animal welfare board at the Ministry of Tübingen confirmed that a permit was not necessary, as the test was not considered to impose any pain, damage or suffering on the animals. The process was registered by the animal welfare board of Nuertingen-Geislingen-University under the number 2021_03_14.0.21. Ethical review and approval for human participation were waived, as no personal data of any test person were used in this study. All persons and animal owners agreed to the use and publication of their anonymous data.

### 2.6. Data Collection and Analysis

#### 2.6.1. Data Preparation

The data from the experimental sheets were inserted into digital data sheets using *MS Excel* (Microsoft Corporation, Washington, DC, USA). Along with indices for motor laterality in the forelimb test, and sensory laterality in the person test and the object test, we recorded the factors of sex, breed, age and training frequency of the horse, as well as the side from which the horse was led. The laterality indices (LIs) were calculated for each horse using the formular (R − L)/(R + L + A), where R is defined as the number of right responses, L as the number of left responses and A as unclear lateral behaviour [23]. The LI scale ranged from −1 to 1, with values below 0 indicating left laterality and values above 0 right laterality. If the LI was exactly 0, the horse was considered bilateral. The LIs were calculated for each horse and for each experiment [23,25], and LIs were calculated separately for the “with rider” and “without rider” (E1) conditions.

#### 2.6.2. Data Analysis

The data were analysed using the software *R statistic* (R Development Core Team, Boston, MA, USA, 2021) and the package *R commander* (Appendix A). Prior to the correlation analyses, the data were tested for normal distribution using a Shapiro–Wilk normality test. Not all parameters were normally distributed (Shapiro–Wilk *p* < 0.05).

Generalised linear models, one for each test situation, were used to analyse differences in the strength of laterality between the “without rider” and “with rider” conditions (formular = LI ~ without/with.rider + training frequency + side.of.leading + sex + breed + age, family = gaussian(identity); Appendix A). All factors were fixed factors set by the experiment. To find the best fit of the model, factors with the least significant *p*-value were excluded stepwise, but only when the exclusion resulted in a decrease in the Akaike information criterion (AIC) or an insignificant increase in the AIC (*p* > 0.05, comparison of the two GLMs—models that resulted in an increase in AIC by using ANOVA; Appendix A) [26]. Only the best fit of the model is shown in the results. The significance level was set at 0.05 for all tests. All tests were two-tailed.

## 3. Results

### 3.1. General Results

In the motor laterality foreleg preference test, the LI median of all tested horses without and with a rider (*n* = 23) was 0.0 (without rider: Min = −0.6; Max = 0.6, with rider: Min = −0.8; Max = 1.0), and, therefore, the horses in this study show equal numbers of left- and right-sided individuals. In the sensory laterality object test without a rider, the LI median was −0.3 (Min = −1.0; Max = 1.0), and, with a rider, it was 0.3 (Min = −1.0; Max = 1.0). In the sensory laterality person test without and with a rider, an LI median of 0.2 (without and with rider: Min = −1.0; Max = 1.0) was noted.

The factors sex, breed, age and the training frequency of the horse, as well as the side from which the horse was led, did not have a significant effect on the models (GLMs: all *p* > 0.05) and were excluded from further analysis. The data of the ponies differed from those of other breeds, but only in the sensory laterality test (person) test.

### 3.2. Influence of a Rider on the Direction of Laterality

No significant differences were found in the direction of laterality between the “without rider” and “with rider” conditions in the motor laterality test (GLM: *n* = 23, t = −0.42, *p* = 0.7), in the sensory laterality measured by using a novel object test (GLM: *n* = 23, t = 0.95, *p* = 0.3) or in the sensory laterality measured by using a person test (GLM: *n* = 23, t = 0.08, *p* = 0.9).

### 3.3. Influence of a Rider on the Strength of Laterality

There was a significant difference in the strength of motor laterality between the “without rider” and “with rider” conditions (GLM: *n* = 23, t = 2.68, *p* = 0.01, Figure 5). Horses showed a stronger motor laterality when carrying a rider. No significant differences were found in the strength of sensory laterality in the novel object test (GLM: *n* = 23, t = 0.29, *p* = 0.8) or in the person test (GLM: *n* = 23, t = −0.19, *p* = 0.8), but the strength of sensory laterality in the person test was significantly lower in ponies than in other breeds (GLM: *n* = 23, t = −2.8, *p* = 0.008, Figure 6).

## 4. Discussion

Three key findings emerged from this preliminary study: Firstly, motor laterality significantly increased in strength when the horses had a rider on their back, but it did not change in direction; secondly, sensory laterality did not change in response to a rider; and thirdly, the ponies were less strongly lateralised than the horses in the person test.

With regard to motor laterality, this result is in line with the findings of Murphy et al. [15], who found that mares showed stronger motor laterality with a rider than without. This could simply be a function of the person’s weight enhancing motor preferences, but it is possible that other factors that were not in the focus of this preliminary study came into play.

In a comparable experiment, Marr et al. [23] found that horses showed a cognitive bias, with horses stepping over a starting line with the right foreleg first being more optimistic than those starting with the left foreleg. It is therefore possible that a known rider increases the expression of the existing state of mind. It is also possible that a horse’s natural crookedness, meaning the morphological asymmetry of the horse [4], has an effect in that the weight of the person may strengthen the predisposition to one side or the other [27]. Furthermore, it cannot be excluded that, although the riders tried to provide minimal aids, the natural laterality of the riders themselves [28], or a slight asymmetry in their seats or balance, may have influenced the horses’ motor laterality.

Interestingly, this experiment did not find a particular bias for one side in motor laterality. This is in contrast with some studies [17,18] that found left foreleg preference in Thoroughbreds and performance horses, but it is consistent with others [25,29] that found no population-level lateral preference in feral and Prewalski’s horses. This suggests that the expression of laterality for one particular side may be dependent upon breed and/or living conditions. For example, Zucca et al. [30] reported changes in motor laterality in donkeys in response to restricted space. As Thoroughbred and performance horses are more likely to have restrictions in their basic needs than leisure horses or horses living under natural conditions [31], this could account for this apparent discrepancy.

However, while motor laterality increased when carrying a rider, sensory laterality remained unchanged. Other studies, including those of Marr et al. [22] and Larose et al. [32], have shown sensory laterality to be connected with stress and emotionality, so the stability of sensory laterality in this study suggests that a largely passive rider does not have a negative impact on a horse’s mental state and stress level. This supports the often recommended training practices of taking breaks in training, in which the rider reduces aids, and of using a rider who remains as passive as possible when a horse is ridden for the first time, with a trainer on the ground initiating movement to reduce the chances of a bad reaction from the horse [33,34,35]. However, it should be noted that all the horses in this experiment were experienced and familiar with their riders, and stress may well be more apparent in young and inexperienced horses or with unfamiliar riders.

The third finding of the horses being significantly more strongly lateralised than the ponies in the sensory laterality person test is consistent with that of Farmer et al. [36], in which ponies also showed the trend of being less strongly lateralised than horses in interactions with conspecifics. There are many reasons why this may be the case. Training has been shown to be a factor in sensory laterality [22,37], and while horses are generally handled by adults, and handled predominantly from the left, ponies are more often handled by children who are themselves less strongly lateralised [38] and, therefore, more bilateral in their interactions with ponies. Additionally, sensory laterality has been shown to vary between breeds, with less temperamental breeds showing weaker lateral bias [18,32]. As ponies were traditionally used for mining and load carrying, and now for children, their weaker sensory laterality may reflect the calm temperament required for these types of work. However, as there were only five ponies in this study, further research would be needed to investigate this.

It could be argued that the person leading the horse in the experiments may have had an effect on the horses’ laterality, but since the horses were led in all experiments with and without a rider and the leading side was randomised, this effect can be excluded.

In summary, these preliminary findings show that carrying a rider did, indeed, influence a horse’s motor laterality [27], but it did not affect sensory laterality under the test conditions. As sensory laterality has been linked to stress and emotionality [6,19,23], this has important implications for stress reduction and welfare in training, and it should be further evaluated in follow-up studies whether riding as passively as possible, on a loose rein, reduces the strength of sensory laterality and stress in a training context. Such research may also distinguish between horses of different sexes, breeds, ages, temperaments and training [15,16,17,18,19,39] and may include behavioural and physiological parameters to evaluate stress [40].

## 5. Conclusions

Carrying a rider significantly increased the horses’ motor laterality but did not affect sensory laterality in this study. This suggests that a rider who is as passive as possible has little or no adverse effect on a horse’s stress levels and mental state. The horses’ sex, age, training frequency and leading side did not influence laterality. However, ponies showed weaker sensory laterality than other breeds in the sensory laterality (person) test.

## Figures and Tables

**Figure 1 animals-12-00992-f001:**
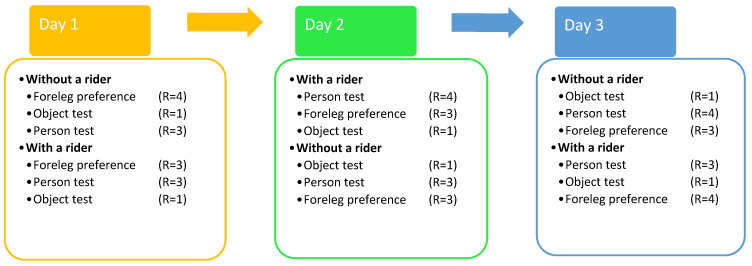
Example of the experimental timeline for each horse. The figure presents a visualisation of the experimental procedure over the three days, showing an example of the order and the number of repetitions (R) of the experiments conducted with a rider (E1) and without (E1) for one particular horse within a day. The single experiments are named as follows: motor laterality foreleg preference test (foreleg preference); sensory laterality person test (person test); and sensory laterality object test (object test). The order of the experiments was chosen by random for each particular test horse so that further horses would complete the same tests but in a different order.

**Figure 2 animals-12-00992-f002:**
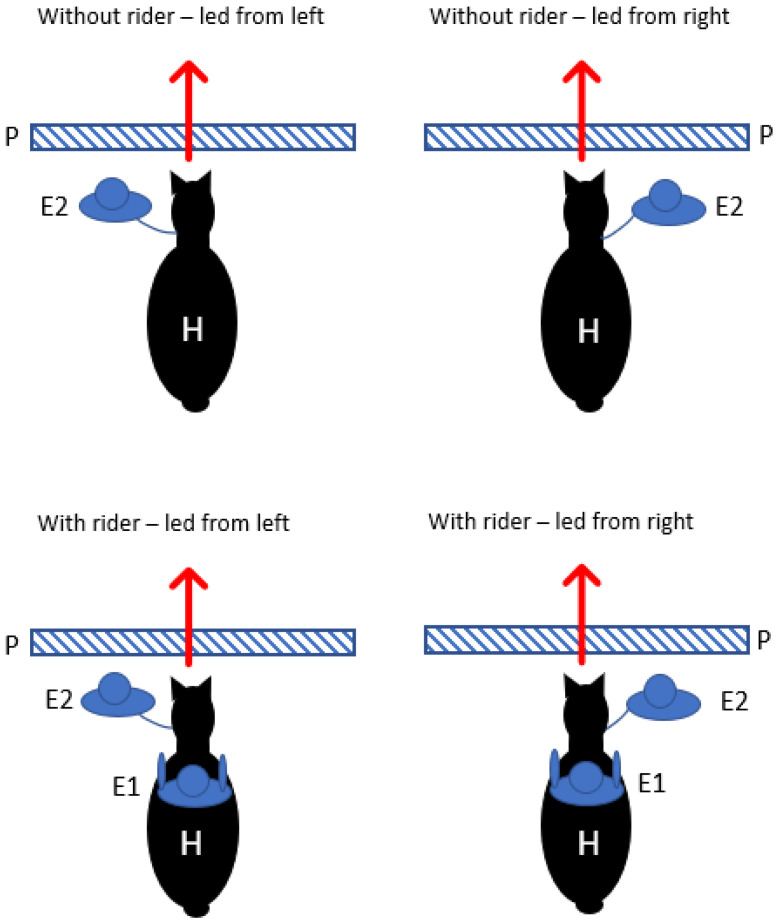
Setup for motor laterality foreleg preference test. The four variations of the foreleg preference test showing the horse (H) being led by Experimenter 2 (E2) from the left or right side, either with or without a rider (E1). The arrow indicates the direction of the horse (H) and Experimenter 2 (E2) stepping over the pole (P) on the ground.

**Figure 3 animals-12-00992-f003:**
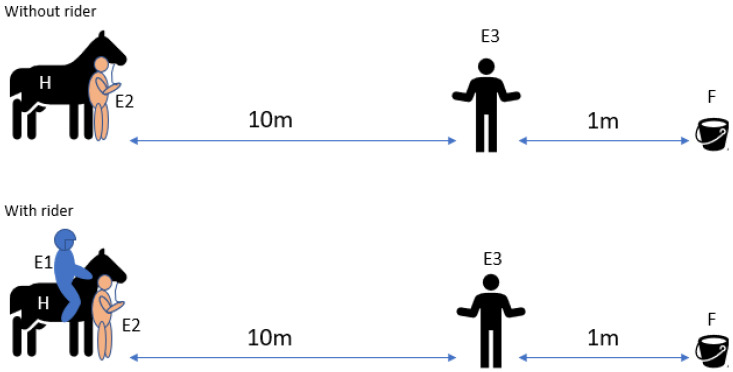
Setup of sensory laterality person test. The setup of the sensory laterality person test shows the horse (H) held by Experimenter 2 (E2) with and without a rider (E1). Experimenter 3 (E3) is positioned in between the horse (H) and the feeding bucket (F), approximately ten metres distance from the horse and one metre in front of the feeding bucket (F).

**Figure 4 animals-12-00992-f004:**
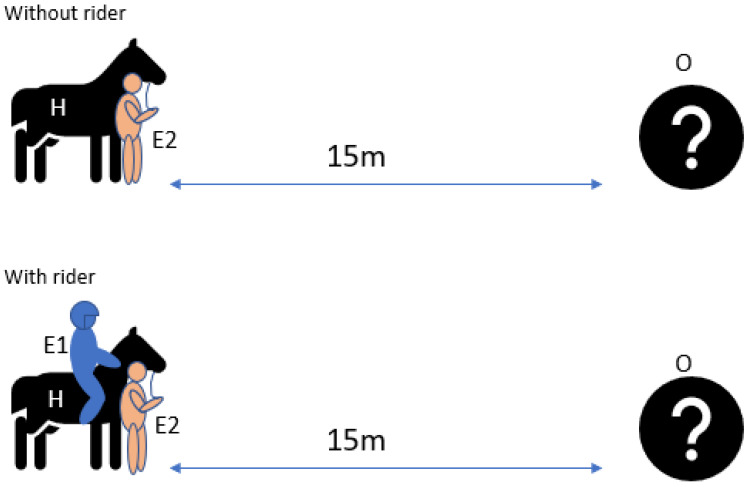
Setup of sensory laterality object test. The setup of the sensory laterality object test shows Experimenter 2 (E2) leading the horse (H) with and without a rider (E1) towards the object (O) from a 15 m distance.

**Figure 5 animals-12-00992-f005:**
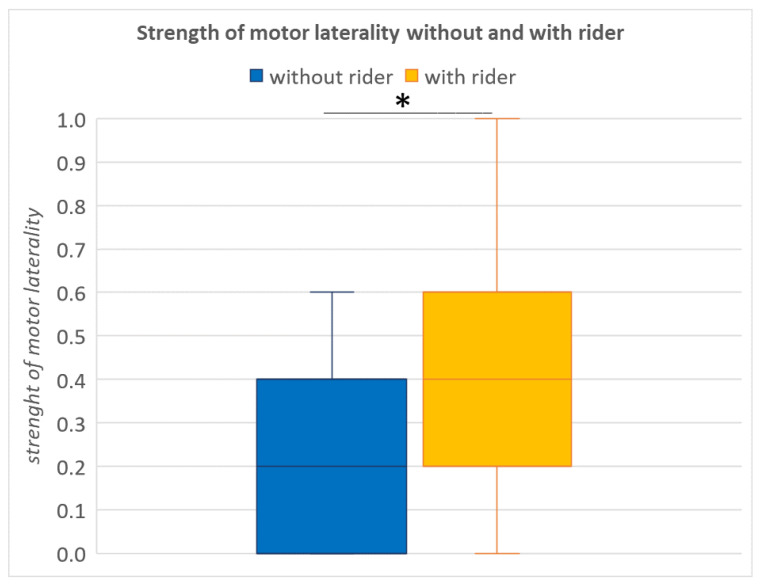
Strength of motor laterality increased significantly when a rider was on the horse (GLM: *n* = 23, t = 2.68, *p* = 0.01). The * visualises a statistical significance between *p* = 0.05 and *p* = 0.01.

**Figure 6 animals-12-00992-f006:**
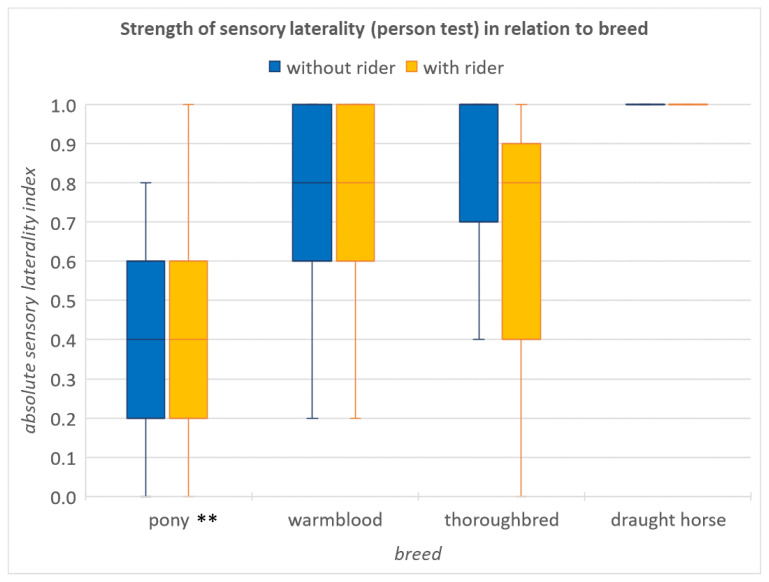
Strength of sensory laterality (person test) with and without a rider in relation to breed. Ponies were significantly less strongly lateralised than horses (GLM: *n* = 23, t = −2.8, *p* = 0.008). The ** visualise a statistical significance between *p* = 0.01 and *p* = 0.001.

## Data Availability

All raw data are provided in the manuscript and in the Appendix A. Reasonable requests for access to experimental videos can be obtained from the corresponding author.

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
