# Peer review of "Does Carrying a Rider Change Motor and Sensory Laterality in Horses?"

_animals, 2022, doi:10.3390/ani12080992_

Round 1

Reviewer 1 Report

The manuscript entitled « Influence of a passive rider on motor and sensory laterality in horses” is an interesting small study of both sensory and motor laterality in horses. The authors explain their methods in a great amount of detail. The language is adequate. The results were clearly presented and concise: in 23 horses, the presence of a passive rider affected the motor, but not sensory laterality.

However, I have several concerns about this study, particularly with regards to the sample size and the number of experimenters. Considering the extent to which the authors have described the study design, would it not make sense to include this design in the GLM? If the experimenters 1, 2 and 3 were testing multiple horses, the effect of the experimenter could be accounted for in the model (or rather, this potential effect should be tested for). Otherwise, the material and method section can probably be thinned out a bit.

The sample size is quite limited to talk about a population level. Furthermore, it is not clear how previous experience of bilateral handling affected the results. The age and sex distributions were not ideal.

I would recommend this article to be considered after major revisions.

Detailed comments:

Abstract:

Generally, in the simple summary, no statistical results are needed, but in the abstract, you should describe the statistical methods and results.

Page 1, line 31: do not use the extrapolation “population level”. Your sample of 23 horses does not represent a larger population of horses.

Introduction:

Page 2, line 53-56: clumsy writing. “For this, …” Maybe use “therefore”?

Page 2, line 56: remove “and”

Page 2, line 60: “surroundings” instead of “sourroundings”

Page 2, line 69-74: it is not clear to me why a left-shift in sensory laterality is associated with a dominance of the right brain hemisphere. Please elaborate in text.

Page 2, line 78: “sensory” instead of “senory”.

Page 2, line 79: up to now, you have not explained that laterality evolves on a spectrum. This should be added in the introduction.

Material and methods:

Page 3, line 102: were there horses that were never trained under the saddle, e.g. the ponies? This is not clear here and would have immense implications on the study.

Page 3, line 116: I do not understand N=11? 23 horses were ridden by their own rider, but but several horses shared the same rider?

Page 4, line 141-142: this information is not necessary

Page 9, line 270-272: how did you handle your repeated measures? If I understood correctly, you had 3 measures per test. Therefore, you would need to correct for this as well, for example by including horse as a random factor. Similarly, you have described different experimenters in detail, but have not included them in the statistical models, why?

Results:

Page 10, lines 284, 286 and 287: do not use the expression “population level”. 23 horses of different breeds from 10 different stables do not constitute a population.

Page 10, lines 304-308: it is not clear here whether the ponies were also ridden. Could the effect of sensory laterality in the person test be due to differences in perceptions due to height? Ponies and draft horses are likely to have different fields of visions of humans due to height differences.

Page 11: the graphs should be the same size. As you have so few tests, you should also represent all your tests, regardless of significance, although you can arrange them by order of significance.

Discussion:

Page 12, line 338: what you describe is not a population.

Page 13, line 364: “strongly” not “stongly”

Page 13, line 367: remove “in”

Reviewer 2 Report

Animals 1632543

The paper is potentially interesting, however to much information is not clear. First of all the authors should underline the reason for their investigations –it is not clear from the introduction (and the rest of the paper).

Tittle and introduction:

The title and the other parts of the paper should be re-written as the rider is not passive – he/she uses aids. The aims of the study are not clear – it is not obvious what is direction and how strength of laterality was expressed. It should be more clear even from the introduction.

Material and methods:

There are many factors that were not taken into consideration – stable, time of work only partially (times of work in the weak but not every day work time), handling habituation.

The age of horses is very different. Probably it would be better to exclude the oldest or the youngest, as the investigated group is not equal. The age affect could be connected with so much difference especially if there are many, not investigated influences.

The authors should think again about breed differences – perhaps exclusion of two cold bloods could help (perhaps they are also the youngest/oldest?). Such mixture of bred/age/training experience does not help in correct analysis of the data.

The experiments and tests are well written, however tests should be explained before examiners, as it would help to understand better their role. There are many not clear sentences or not truth sentences in the description – for example :

L 127 – loose or equal length reins of “passive” rider is not the same

L125-127 – “usual leg aids” is not a passive riding – such description is used many times

L 122 – rider rode the horse one round in each direction with a random change of rein as a passive riding? Not possible. The riders may use their weight/balance without using reins. So it is not passive riding.

L 130- experience of leading person is also important

L 137- posture/position of E3 should be described better

L 159 –perhaps these lateral responses should be described also?

Figure 1 – should state alone without the text – and it is not clear at all. Example? Does it mean that every horse can has different one? It looks like 15 test for horse every day – that is not in according with the text of the paper

L 177-178 it is not clear if persons working with horses (not ridding) were the same for all horses or changed? If they were different it should be in the statistical model.

L 184 – once again it is not passive riding

L 185-186 – not in accordance with the figure 1.

Perhaps the leading person and or standing person should be in the statistical model too (as random effects?) or were they the same for all?

L 235-236 – other objects you mean?

L 251 - Data   - first the measured traits should be described and in a better way like it is in this part below. You should clearly state what is “direction of laterality” and what is “strength of laterality”. That is not detailed describe and underlined. Then in the result part your results are confusing.

It seems that not all possible (and controlled ) environmental factors were taken into account. You have put into the model side of leading but the leading habituation is not taken into account. You have taken into account frequency of training but nothing on its intensity is written (1-2 h per day? or more?). That can be of great importance also. There are so many factors not taken into account and so uneven group of horses investigated that I would try again to construct more proper investigation group. Perhaps you could exclude some horses without too much loss of the information? In the way it is now it seems that you do not cover in the model the variability you have.

Data analysis – you write that you have not normally distributed parameters (L 269) and you use statistic method-  analysis of variance (?) for normally distributed data? You have even not written what measure/trait is not normally distributed. Please explain.

The statistical method and the model should be described[ better (L-270-278). What is it “family-gaussian identity”? L 273

AIC criterion  (l276) is usually used by mixed models –and you write that your factors were all fixed (L273)?

Without answers to these questions it is difficult to evaluate your results.

However you write – that you have excluded all of your factors – L290 because they were not significant; but the ponies differ in lateralization – L 360?

L 237- perhaps you should underline it in the title that it is preliminary research, or think about sending this paper as the preliminary work (case report). Otherwise  the investigated group of horses seems unequal for clear results.

L 331 – optimistic horse? I think you should try to find better expression, even if it is the original one. L 332 – explanation ok

L 347 – it does not seem your result

Discussion – lines 360 and 367 are not compatible. Please checked again.

L 366 – please underlined clearly that they may be just less trained. Equal laterality, balance  is trained – see Meij and Meij 1980.

Meij, HS* &Meij, JCP. "Functional asymmetry in the motor system of the horse." South African Journal of Science 76.12 (1980): 552.

Conclusions – it looks like the last part of discussion with many citations, and not like your own conclusions. Please rewrite. Underline your own result and its importance.

Round 2

Reviewer 1 Report

What could be addressed was addressed, thank you for that. I have found some minor spelling errors in the added text, please correct before publication.

Page 2, line 54: that is to say

Page 3, line 89: are used

Page 3, line 91: Does

Page 10, line 307: of this study

Page 10, line 314-315: rephrase: “The data of the ponies differed from other breeds, but only in the sensory laterality test (person test).”

Author Response

Responses to reviewer 1 (Round 2)

Reviewer 1:

  • Thank you very much for pointing out the typos, we corrected all the mistakes.

Reviewer 2 Report

Dear Authors,

Thank you for corrections and explanations.  I have some remarks to the new version of the paper:

  1. clarity of test number and description of the figure 1 – your example of the experiment can be included in the text of the paper – as it was not quite clear from the text also.

Please add some sentences about the possible influence of the numbers of tests per day per horse into the discussion – the number of tests seems too much for one horse per day. The tests are not very intensive from the physical point of view, however the possibility  of the horse perception was surely lowered by 10th or 14th test. That should be taken into account at least  at the discussion (probably by the statistical model?). The ethical statement is not quite clear.

  1. statistical analysis – please use direct citation of your statistical method used. The readers have to be informed where can they find statistical explanation on. The name of program is not enough. Probably the one you cited in the answers to review?
  2. conclusions – the conclusions are better now, however they can be better. They are too short. Please add the information about the effects that were not statistically significant in your preliminary research.
  3. Please change the word "passive" also on the tittle (it is changed  though all the papier).

Author Response

Responses to reviewer 2 (Round 2)

Reviewer 2:

  1. clarity of test number and description of the figure 1 – your example of the experiment can be included in the text of the paper – as it was not quite clear from the text also.
  • Thank you, we tried to clarify the method in ll. 164-165. Apart from that, we find it useful to have figure one as a visual example, since otherwise it would be hard to explain.

Please add some sentences about the possible influence of the numbers of tests per day per horse into the discussion – the number of tests seems too much for one horse per day. The tests are not very intensive from the physical point of view, however the possibility  of the horse perception was surely lowered by 10th or 14th test. That should be taken into account at least  at the discussion (probably by the statistical model?). The ethical statement is not quite clear.

  • The ethical statement section states, that due to the responsible institution our experiments did not contain severe animal testing and thus did not require a permit. Our tests were not considered to cause the horse any pain, damage or suffering. The ethical board of the University Nürtingen and the animal welfare board of the ministry for agriculture gave this study a thorough examination.
  • Since only maximal four repetitions of one experiment were conducted in a row, it is very unlikely that the horses’ perception suffered. The tests were changed frequently and the duration of the experiments was never very long. The number of repetitions conducted in the present study are much lower than have been conducted in many other studies.
  • Furthermore, as the order of the three tests were randomised serial effects on the behaviour of the horses and therefore effects of the results of the particular tests can be excluded. If the horses had suffered attention loss, this would have affected all tests, and the overall result of the three tests would still be comparable.
  • It does not need to be discussed in further detail, as the procedure of randomising the test and the number of tests is common practise.
  1. statistical analysis – please use direct citation of your statistical method used. The readers have to be informed where can they find statistical explanation on. The name of program is not enough. Probably the one you cited in the answers to review?
  • Thank you for your remark. We added the citation you mentioned for explaining the AIC.

  1. conclusions – the conclusions are better now, however they can be better. They are too short. Please add the information about the effects that were not statistically significant in your preliminary research.
  • Thank you, you are right, they were quite short. We added another sentence about the factors that were tested.
  •  
  1. Please change the word "passive" also on the tittle (it is changed though all the papier)."
  • Thank you very much for pointing us to it, we did so! The system still showed the old title, in the new one “passive” was deleted already.
